# Computational Selectivity Assessment of Protease Inhibitors against SARS-CoV-2

**DOI:** 10.3390/ijms22042065

**Published:** 2021-02-19

**Authors:** André Fischer, Manuel Sellner, Karolina Mitusińska, Maria Bzówka, Markus A. Lill, Artur Góra, Martin Smieško

**Affiliations:** 1Computational Pharmacy, Departement of Pharmaceutical Sciences, University of Basel, 4056 Basel, Switzerland; and.fischer@unibas.ch (A.F.); manuel.sellner@unibas.ch (M.S.); 2Tunneling Group, Biotechnology Centre, ul. Krzywoustego 8, Silesian University of Technology, 44-100 Gliwice, Poland; k.mitusinska@tunnelinggroup.pl (K.M.); m.bzowka@tunnelinggroup.pl (M.B.)

**Keywords:** coronavirus, SARS, protease, selectivity, structure-based design

## Abstract

The pandemic of the severe acute respiratory syndrome coronavirus 2 (SARS-CoV-2) poses a serious global health threat. Since no specific therapeutics are available, researchers around the world screened compounds to inhibit various molecular targets of SARS-CoV-2 including its main protease (M^pro^) essential for viral replication. Due to the high urgency of these discovery efforts, off-target binding, which is one of the major reasons for drug-induced toxicity and safety-related drug attrition, was neglected. Here, we used molecular docking, toxicity profiling, and multiple molecular dynamics (MD) protocols to assess the selectivity of 33 reported non-covalent inhibitors of SARS-CoV-2 M^pro^ against eight proteases and 16 anti-targets. The panel of proteases included SARS-CoV M^pro^, cathepsin G, caspase-3, ubiquitin carboxy-terminal hydrolase L1 (UCHL1), thrombin, factor Xa, chymase, and prostasin. Several of the assessed compounds presented considerable off-target binding towards the panel of proteases, as well as the selected anti-targets. Our results further suggest a high risk of off-target binding to chymase and cathepsin G. Thus, in future discovery projects, experimental selectivity assessment should be directed toward these proteases. A systematic selectivity assessment of SARS-CoV-2 M^pro^ inhibitors, as we report it, was not previously conducted.

## 1. Introduction

In late 2019, a novel coronavirus termed SARS-CoV-2 emerged and spread around the world causing coronavirus disease 2019 (COVID-19). Until today, over 62 million cases were reported accounting for over a 1.46 million of fatalities (as of 1 December 2020) [1]. While pharmaceutical interventions primarily remained symptomatic, multiple clinical trials are investigating novel treatments, mainly based on drug repurposing [2,3]. Thus, the treatment of this infection with specific drugs constitutes an urgent and unmet medical need. In the pharmaceutical treatment of viral infections such as human immunodeficiency virus and hepatitis C virus, the inhibition of viral proteases is a successfully applied strategy. Consequently, many computational and experimental efforts were directed toward targeting the main protease (M^pro^) of SARS-CoV-2 with small molecules leading to the discovery of multiple promising candidates [4,5,6,7,8,9]. Due to the high urgency and the competitive scientific field, off-target binding was rarely considered in the latest discovery projects. However, a large share of drug attrition in clinical trials, especially regarding compound safety, can be traced back to low target specificity and off-target binding [10,11,12]. To avoid the large cost associated with late stage drug failure, early off-target profiling, especially with comparably economical computational methods, offers an attractive strategy [12,13,14]. On the other hand, binding to multiple targets can be beneficial in specific cases, such as pan inhibition of the viral proteases of SARS-CoV-2 and SARS-CoV [15]. Similarly, the concurrent inhibition of the coagulation protein factor Xa and SARS-CoV-2 M^pro^ constitutes another example for a potentially synergistic effect of multi-target binding as COVID-19 infection is associated with life-threatening coagulopathies that can be treated with anticoagulants [16,17]. Computational methods such as molecular docking and specialized molecular dynamics (MD) protocols can be exploited to explore the selectivity of small-molecule compounds, as it was evidenced for various targets. For example, we previously applied docking combined with cosolvent MD simulations and determination of hydration hot-spots to investigate the selectivity of allosteric inhibitors against eight nuclear receptors to support targeted experimental profiling of novel compounds [18]. In general, it was discussed that cosolvent MD simulations can provide valuable insights for the development of potent and selective compounds [19,20]. Although multiple studies focused on the selectivity assessment among kinases using computational methods [21,22], there were only minor efforts to establish selectivity factors for small-molecules that bind to proteases [23,24].

Here, we examined the selectivity of 33 experimentally confirmed non-covalent SARS-CoV-2 M^pro^ inhibitors against eight different proteases including SARS-CoV M^pro^, factor Xa, cathepsin G, caspase-3, prostasin, thrombin, ubiquitin carboxy-terminal hydrolase L1 (UCHL1), and chymase by molecular docking (Table 1). The  proteases were selected based on a structural similarity search in the NCBI database, as well as considerations regarding their pharmacological relevance. First, we compared the active sites regarding pharmacophores and electrostatic potential. Furthermore, we performed classical as well as cosolvent MD simulations to identify water and small-molecule hot-spots of the respective active sites offering explanations for compound selectivity. Based on our results, experimental selectivity profiling in future discovery projects can be directed toward targets with an inherent high liability for off-target binding. Up to this date, such a comprehensive evaluation of off-target binding of SARS-CoV-2 M^pro^ was not previously conducted and is of high importance to support antiviral drug development.

## 2. Results and Discussion

### 2.1. Sequence and Active Site Comparison

We selected eight proteases to assess off-target binding and to structurally compare them to SARS-CoV-2 M^pro^. The  proteases were selected based on their catalytic residues, sequence and structural similarity, availability of structural information, as well as their pharmacological and physiological relevance. Except for SARS-CoV M^pro^, they exhibited different overall folds and showed low global sequence similarity to SARS-CoV-2 M^pro^ (Table 2). Furthermore, the volumes of the active site cavity (Appendix A) of the SARS-CoV-2 M^pro^ was the smallest among all analyzed proteases, and regarding absolute values, most similar to chymase as opposed to SARS-CoV M^pro^. However, when we compared the active sites of all analyzed proteases, they presented a remarkable degree of similarity. The  root mean-square deviation (RMSD) of their catalytic residues did not exceed 2 Å, except for UCHL1 with 2.8 Å. Furthermore, when using FuzCav [25] to determine the similarity of the active site pockets of the panel of proteases, we observed that not only their catalytic residues, but also the complete active sites are similar relative to SARS-CoV-2 M^pro^ (Appendix A). Actives sites exceeding a similarity value of 0.16 can be regarded as similar [25]. We also compared the electrostatic potentials of the binding cavities of all analyzed proteases using PIPSA [26] software. The  electrostatic potentials were compared quantitatively by calculating the Hodgkin similarity index (Table 2, Appendix A). Using the Hodgkin similarity index it is possible to determine the correlation between the potentials of the analysed proteases (+1 indicates that potentials are identical, 0 indicates that potentials are fully uncorrelated, and -1 indicates that potentials are anti-correlated). Three out of eight proteases (SARS-CoV M^pro^, prostasin, and thrombin) presented a positive correlation, whereas the remaining ones indicated an anti-correlation in relation to the SARS-CoV-2 M^pro^ binding cavity. In the case of UCHL1, the high anti-correlation was caused by a specific binding site spot with reversed distribution of electrostatic potentials. Prostasin, thrombin and SARS-CoV M^pro^ also showed high similarity relative to the active site of SARS-CoV-2 M^pro^ using FuzCav. Thus, even though the substrate specificity of the panel of proteases is diverse [27], their exceptionally similar active sites compared to SARS-CoV-2 may suggest a potential for off-target binding of small-molecules.

### 2.2. Hydration and Small-Molecule Hot-Spots

In the next step, we further characterized and compared the selected proteases according to their hydration and small-molecule hot-spots by using different molecular probes including water, acetonitrile, isopropanol, and pyridine. The  use of specific functional groups represented by the different organic probes associating with the active sites can be used to fine-tune the selectivity profile of protease inhibitors [18,28]. Similarly, selectively targeting hydration sites occurring in one protein, but not in an anti-target, offers potential to be exploited in structure-based design. It should however be mentioned that whether the displacement of water molecules is favorable or not depends on the thermodynamic profile of the respective hydration site [29], which was not assessed in this work. The  comparison of the hot-spots in the vicinity of the active site residues revealed distinct similarities (Figure 1).

For SARS-CoV-2 M^pro^, we identified two hydration sites located in the vicinity of H41, as well as a small-molecule hot-spot for acetonitrile molecules at the same location (Figure 2). While we could not detect a hydration site in the vicinity of C141, association of pyridine and isopropanol was detected. In the case of SARS-CoV M^pro^, however, four hydration sites could be identified. Potentially, the increased flexibility of SARS-CoV M^pro^ allowed for increased solvent accessibility in comparison to SARS-CoV-2 M^pro^ (Appendix A) [30]. Furthermore, the increased magnitude of cosolvent densities in SARS-CoV M^pro^ confirmed this observation, although they mainly occupied the vicinity of H41. Even though one hydration site between the M^pro^s overlapped, there were significant differences between the small-molecule hot-spots of the SARS-CoV proteases, which have to be accounted for in the design of pan inhibitors against the two coronaviruses (Figure 2). For caspase-3, two unique hydration sites were identified in the active site cavity distant from the catalytic residues, which were not observed in the SARS-CoV-2 M^pro^. One of the hydration sites overlapped with the occupancy of multiple organic probes, which sampled a unique region not observed in any other protease besides UCHL1. Three hydration sites were identified in the vicinity of the active site residues in UCHL1. One of the hydration sites was located between three catalytic residues as in caspase-3, but none of the other cysteine proteases. Thus, these proteases are similar, while presenting distinct differences to the M^pro^s despite their shared catalytic mechanism using cysteine for the nucleophilic attack of the substrate (Table 2). In the active site of factor Xa, we detected two hydration sites matching the position in prostasin and chymase indicating that they are conserved. As one of the sites (denoted as site B in Figure 1) could not be observed in the SARS-CoV-2 M^pro^, it might contribute to ligand specificity. Remarkably, the cosolvent densities among factor Xa, cathepsin G, thrombin, and chymase presented a high degree of similarity, with an additional density for acetonitrile compared to SARS-CoV-2 M^pro^. The same region was occupied by pyridine probes. A common density of pyridine in the center of the sites, however, suggested a common preference for hydrophobic or aromatic moieties among the aforementioned enzymes. Comparing the organic probe density of factor Xa to SARS-CoV-2 M^pro^, a common preference for acetonitrile on the distal side of the catalytic histidine could be observed. This could guide the placement of an amphipathic moiety in this region to inhibit both proteases with future antivirals. While cathepsin G and thrombin shared one of the most commonly observed hydration sites, they lack the common site observed in the viral M^pro^s which indicates that the displacement of this water molecule (denoted as site A in Figure 1) could contribute to selective binding. Both thrombin and chymase presented a high number of hydration sites within their active sites sharing the above-mentioned hydration site B not observed in the viral proteases, as well as the previously discussed acetonitrile density near the backbone of the catalytic histidine residue. Thus, the inherent potential for off-target binding of novel antivirals to these targets is small, similar to caspase-3 and UCHL1. Compared to the volume of its active site cavity, the site of chymase seemed highly accessible to the surrounding solvent. Overall, the highest similarity among the proteases regarding hydration sites and small-molecule binding hot-spots, could be observed for factor Xa and chymase. Differences in hydration site locations identified for individual simulations are most probably related to conformational changes of the proteins, as the volumes of the active sites underlined (Appendix A).

### 2.3. Protease Selectivity Assessed by Molecular Docking

Binding modes obtained from molecular docking have been widely used to establish selectivity factors toward different targets [13,18,31]. Here, we compiled compound sets comprising experimentally verified ligands of nine proteases to assess the selectivity of recently reported SARS-CoV-2 M^pro^ inhibitors (Figure 3) by cross-docking them into the respective protein active sites. These known binders were either retrieved from a set of cocrystallized ligands, the PubChem BioAssay database [32], or the literature (Appendix A). First, to ensure accurate pose-prediction of our computational models, we retrieved numerous crystal structures from the Protein Data Bank for each target and cross-docked their cocrystallized ligands, which is a common procedure in virtual screening projects [33]. Based on the obtained RMSD values between predicted and native binding poses, ensembles of protein structures that yielded best-possible pose prediction quality for non-covalent ligands were identified. In the case of unsatisfactory performance of these structures, short MD simulations were performed to enrich the proteins’ structural diversity. These procedures were performed for the Glide standard precision (SP) and smina docking protocol, to address known differences among docking programs. This resulted in excellent docking accuracy for most protein systems which ranged from 75% to 100% of cocrystallized ligands being predicted below an RMSD threshold of 2.5 Å. The  only exception was prostasin, for which we only found an accurate pose for one of the two available ligands (Appendix A). Unfortunately, no non-covalent small molecule was cocrystallized with UCHL1 which prevented us from computing these metrics in this case. In a next step, we evaluated the performance of the selected ensembles to distinguish between known actives and randomly selected decoy molecules based on the Area Under the Curve (AUC) of the Receiver Operator Characteristic (ROC) curves. Considering the best score of each compound against the ensemble, acceptable ROC AUC values between 0.631 and 0.953 were obtained demonstrating the accuracy of our models and procedures in both detecting the actives and predicting bioactive conformations (Appendix A).

After the validation of the docking protocols, we performed the selectivity analysis based on docking scores. In detail, the SARS-CoV-2 M^pro^ inhibitors were docked to every selected protease and their docking scores were compared with those of the native ligands of the respective enzyme. Again, we determined the ROC AUC metrics with the SARS-CoV-2 compounds regarded as decoys. Except for cathepsin G and chymase, the docking calculations with SARS-CoV-2 M^pro^ inhibitors as decoys displayed higher ROC AUC values compared to the docking calculations with randomly selected decoy molecules. This indicates an overall low potential for off-target binding based on this metric. However, in addition to the ROC AUC metric, we depicted the scores in histograms for every target (Figure 4A). The  docking scores of the SARS-CoV-2 compounds were predicted to be comparably high in magnitude with the majority of compounds only scored slightly better than -6.0 kcal/mol, even when docked to SARS-CoV-2 M^pro^ itself. This is not surprising as the experimentally measured affinity for those compounds only reached micromolar IC_50_ values. As already established by the ROC AUC values, cathepsin G presented the highest score overlap between the compound sets suggesting a risk for off-target inhibition of this protease involved in antigen processing. Other proteins with a comparatively high overlap were SARS-CoV M^pro^, UCHL1, and chymase. Further, some inhibitors of caspase-3 presented an overlap with the best-scoring SARS-CoV-2 M^pro^ inhibitors, while the majority of compounds were highly separated. In SARS-CoV M^pro^, thrombin, and chymase several SARS-CoV-2 M^pro^ inhibitors yielded a similar docking score as some of the actives for that target, even though the peaks of the score distribution were well separated. Especially, in the case of thrombin, concurrent binding could benefit COVID-19 patients suffering from coagulopathies such as venous thromboembolism or sepsis-induced coagulopathy [16,17]. In addition to caspase-3, the distribution in factor Xa and prostasin presented a clear separation of the peaks for each compound category indicating a low potential for concurrent binding. In order to obtain more confidence in the results from molecular docking, we used the complexes and subjected them to MD simulations followed by molecular mechanics-generalized Born surface area (MM/GBSA) post-processing. This methodology is considered to be more precise as opposed to docking scores for a multitude of biomolecular systems [34]. Even though the spread of the values was higher using this protocol, the general trends remained highly similar, especially for chymase, cathepsin G, and caspase-3 (Appendix A). There was a slightly higher overlap of the scores for factor Xa and SARS-CoV-M^pro^, which would indicate a higher potential for a compound to hit both targets.

Interestingly, when the actives of each target were docked to SARS-CoV-2 M^pro^, prostasin inhibitors yielded better docking scores compared to the native ligands (Appendix A). We noticed similar, but less pronounced trends for inhibitors of thrombin, factor Xa, chymase, and caspase-3, while cathepsin G, SARS-CoV M^pro^, and UCHL1 compounds presented nearly identical maxima of their docking scores compared to SARS-CoV-2 M^pro^ inhibitors. In conclusion, the distribution of the scores indicate promiscuity toward chymase, UCHL1, and especially cathepsin G, even though the majority of SARS-CoV-2 M^pro^ inhibitors presented inferior binding scores towards all assessed proteins. Notably, empirical scoring functions, as they were used in this project, have a known degree of inaccuracy, and thus, the absolute numbers should be regarded only as trends.

To acquire structural insights into the selectivity factors of each protease, we visualized binding modes of ligands presenting either low or high binding affinities for each target. According to these complexes, compounds intended to inhibit SARS-CoV-2 M^pro^ should present π-π stacking with the catalytic residue H41 as well as high complementarity with the available subpockets of the active site (Figure 4B). To achieve potent and selective interaction with cathepsin G, the binding modes suggest a salt bridge to K192 or H57 as well as a deeply buried hydrophobic moiety to be optimal (Figure 4C). Similarly, ionic interactions, especially if they were buried, seemed to play a role for selective binding toward thrombin (Figure 4D), caspase-3 (Figure 5A), and chymase (Figure 5B). Interestingly, compounds hitting the presumably desired off-target factor Xa also strongly relied on shape complementarity as for SARS-CoV-2 M^pro^ (Figure 5C), which might explain the concurrent binding to these targets, as we have previously detected in a virtual screening project [4]. Two-dimensional (2D) depictions of all discussed binding modes are presented in the Appendix A.

### 2.4. Toxicity Profiling

As mentioned in the introduction, affinity toward anti-targets is frequently responsible for drug attrition [12]. To establish toxicologically relevant binding profiles of drug candidates our lab has developed the VirtualToxLab (VTL) evaluating their interaction with 16 anti-targets relevant for endocrine disruption, cardiac adverse effects, and extensive or undesired metabolism [13]. Besides estimates for binding affinities against the anti-targets, the VTL provides a parameter referred to as toxic potential serving as a consensus readout for potential undesired effects of the respective compound. SARS-CoV-2 M^pro^ inhibitors with toxic potential significantly higher than 0.5 (Appendix A) included compounds **1** (*(R)*-beperidil), **2** (*(S)*-beperidil), **29**, **31** (nelfinavir), and **32** (lopinavir). While compounds **1** and **2** were predicted to interact with multiple nuclear receptors, the hERG channel, and various cytochromes, compounds **29**, **31**, and **32** presented affinity toward a more narrow spectrum of anti-targets (Figure 5D). Compound **29**, for example, almost exclusively bound to nuclear receptors, resulting in a high estimated risk for endocrine disruption [13]. A common feature of compounds **29**, **31**, and **32** was their prediction as hERG binders. The  hERG potassium channel is one of the most frequently tested anti-targets in drug development due to its involvement in fatal arrhythmias [12,13,35]. The  results regarding the HIV protease inhibitors nelfinavir and lopinavir included in our study confirmed the predictive power of the VTL protocol, as in vitro experiments evidenced their hERG inhibition [35]. Thus, bepiridil (compounds **1** and **2**) displaying a strong interaction for hERG might be at risk to cause cardiac arrhythmia. A large share of the reported SARS-CoV-2 M^pro^ inhibitors have comparably low molecular weights with 26 of 33 reported compounds below 300 g/mol (Appendix A). Since such fragment-like compounds frequently display low specificity [36], the expansion of these scaffolds might generally decrease their potential for off-target binding and at the same time improve their moderate potency.

### 2.5. Selectivity from Different Perspectives

We analyzed the selectivity of nine proteases from different perspectives including sequence and active site similarity, the location of hydration hot-spots and preference for certain chemical probes, as well as molecular docking and toxicological profiling. At the first sight, the low sequence similarity among the proteases (Table 1), the different cleavage sites, as well as the different volumes of the active sites may suggest a low risk of off-target binding. However, all investigated off-target proteins showed a considerable active site similarity based on 3D fingerprints and the positioning of catalytic residues (Table 2). Based on these parameters, prostasin and factor Xa were the most similar proteases compared to SARS-CoV-2 M^pro^. In six analyzed proteases, a hydration site indicated by water hot-spot was identified near the histidine (denoted as site A). Only for cathepsin G, caspase-3, and thrombin, we could not identify this hydration site. Thus, the displacement of this water molecule would not add any ligand selectivity in this regard. Further, for five of the nine proteases, another hydration site was identified, near the serine/cysteine residues in factor Xa, cathepsin G, prostasin, thrombin, and chymase. Regarding the observed cosolvent densities, we detected a density of acetonitrile in factor Xa, cathepsin G, UCHL1, thrombin, and chymase, but not SARS-CoV-2 M^pro^, where this region was explored by pyridine. Distinct placement of pharmacophores matching these differences in density could be exploited to improve inhibitor selectivity. The  similarity of the overall densities in factor Xa, cathepsin G, and chymase coupled to the dissimilarity to SARS-CoV-2 M^pro^, indicated low potential for off-target binding toward these proteases. A common density of pyridine in the center of multiple sites including the one of SARS-CoV-2 M^pro^, however, showed the preference of an aromatic or hydrophobic moiety in this region. Caspase-3 and UCHL1 presented the most unique densities indicating a low potential for off-target binding of inhibitors targeting the remaining proteases.

The docking results of 33 non-covalent SARS-CoV-2 M^pro^ suggested an overall high potential for binding to factor Xa, thrombin, and cathepsin G. On the other hand, the distribution of the docking scores indicated a low potential of the 33 compounds for binding toward UCHL1 and caspase-3 (Figure 4A). As the individual enzymes offered different structural factors relevant for potent ligand-protein interaction, their consideration might improve the design of selective inhibitors. Regarding individual compounds, we identified four compounds which had the highest binding affinity toward more than a half of the analyzed proteases: nelfinavir (**31**) [4,37,38], lopinavir (**32**) [39,40,41], pimozide (**33**) [42], and baicalein (**30**) [43] (Appendix A). Similarly, the aforementioned compounds, as well as both stereoisomers of beperidil (**1-2**) were predicted to interact with a large panel of known anti-targets. Especially, interactions with the hERG potassium channel, as it was also observed in laboratory experiments, raised safety concerns for several compounds (Figure 5D). In this regard, nelfinavir (**32**) was not only predicted to interact with other proteases such as factor Xa, but also towards the hERG channel indicating low selectivity of this compound. Interestingly, when we focused on the compound with the highest predicted binding affinity toward a particular enzyme, we could distinguish three groups: one in which nelfinavir binds best (including SARS-CoV M^pro^, UCHL1, thrombin, and chymase), second in which pimozide binds best (including prostasin, factor Xa, and caspase-3), and third in which baicalein binds best (including SARS-CoV-2 M^pro^ and cathepsin). A closer look into the first group of enzymes revealed that none of them share the same cleavage site, moreover both SARS-CoV and SARS-CoV-2 M^pros^ bind best to different compounds (nelfinavir and baicalein, respectively). As we highlighted selectivity from different perspectives, the different metrics are inherently not always consistent for a single target. To conclude, our predictions indicate the highest potential for off-target binding of SARS-CoV-2 M^pro^ inhibitors for factor Xa, SARS-CoV M^pro^, and cathepsin G. Low potential was determined for prostasin, thrombin, and to the largest degree, for caspase-3 (Table 3).

## 3. Materials and Methods

### 3.1. Selection of Proteases

The panel of proteases for this work were selected using the VAST+ tool [44] by using the SARS-CoV-2 M^pro^ structure (PDB ID: 6Y2E) as input. The  VAST+ protocol determines similar macromolecules to a query structure by computing the superposition of three dimensional protein structures relying purely on geometric measures. The  results were filtered to only match only human proteases. The  next criterion was the the reaction mechanism of the selected protease, to ensure a representation of both cysteine and serine proteases in our study. Finally, we examined the availability of crystal structures with cocrystallized ligands leading to the selection of proteases listed in Table 1. The  preprocessing of the structures is given in the Appendix A.

### 3.2. Similarity of Proteins and Active Sites

The sequence identity was determined using FASTA sequences derived from the UniProt database [45] (Appendix A). In the case of both SARS-CoV-2 and SARS-CoV M^pro^, as well as factor Xa and thrombin, we truncated the sequences to cover the entry in the respective crystal structures limiting the analysis to the catalytic unit. The  sequences were aligned with the ClustalW algorithm [46] in the UGENE suite (v34.0) [47]. The  sequence identity was computed based on matches of the respective protein to SARS-CoV-2 M^pro^ in respect to the length of the sequence.

The active sites of the proteases were aligned in PyMOL using the pair_fit command. Each protease was aligned to the reference structure: the SARS-CoV-2 M^pro^ (PDB ID: 6Y2E), fitting the protease catalytic histidine residue with the H41 of the SARS-CoV-2 M^pro^, protease catalytic serine or cysteine residue with the C145 of the SARS-CoV-2 M^pro^, and the protease aspartic acid residue with the catalytic water molecule (ID 582) of the SARS-CoV-2 M^pro^. The  RMSD values were computed according to the superimposition of the catalytic residues in PyMOL.

To determine the similarity of the active sites of the considered off-targets to SARS-CoV-2 M^pro^, we used the FuzCav [25] routine. This routine computes the similarity based on fingerprints for each binding site incorporating pharmacophoric properties from the coordinates of surrounding α-carbon atoms. As input, we selected residues in 5 Å around the cocrystallized ligands in the structures.

To compare the electrostatic interaction properties of the binding cavities, we used the PIPSA [26] software. First, we preprocessed the structures using PDB2PQR tool [48], and we calculated the Adaptive Poisson-Boltzmann Solver (APBS) electrostatics potentials [49] setting the grid spacing to 0.6. Then, we calculated the similarity matrix (Hodgkin index) of the binding cavities from APBS grids. The  binding pocket was set as a sphere with a radius of 12.5 Å around the geometric centre of the catalytic amino acids after superposition.

### 3.3. Cosolvent MD Simulations

The cosolvent MD simulations were conducted with the Mixed Solvent MD workflow of the Desmond (v2019-1) simulation engine [50] with acetonitrile, isopropanol, and pyridine as probe molecules as they are water-miscible and feature a low potential for aggregation. The  concentration of the probe molecules was selected at 5% (by volume) and, if required for system setup, the water buffer parameter was increased from 12.0 to 15.0, as described in the documentation of the workflow. From the above-mentioned protein structures, monomers were retained to reduce the computational cost of the simulations. Furthermore, the ligands were removed from the structures to sample the respective binding sites. The  simulations were performed with the default specifications at a temperature of 300 K and the OPLS_2005 force field in an NPT ensemble. After an equilibration of 15 ns, production runs of each probe were individually executed for 5 ns with 10 replica simulations resulting in a cumulative simulation time of 600 ns per protein.

### 3.4. Classical MD Simulations

The H++ server [51] was used to protonate all proteins structures listed in Appendix A using standard parameters at pH 7.4. The  missing 4-amino-acids-long loop of the 1Q2W SARS-CoV M^pro^ model was added using the corresponding loop of the 6LU7 model (from SARS-CoV-2 M^pro^), and its quality was confirmed by comparison with another crystal structure of SARS-CoV M^pro^ (PDB ID: 2H2Z). Counter ions were added to to neutralize the systems as shown in Appendix A. Water molecules were placed using the combination of 3D-RISM [52] and the Placevent algorithm [53]. AMBER 18 LEaP [54] was used to immerse models in a truncated octahedral box with 12 Å radius of TIP3P water molecules and prepare the systems for simulation using the ff14SB force field [55]. The  number of added water molecules is shown in Appendix A. The  PMEMD CUDA package of AMBER 18 software [54] was used to run 10 replicas of 50 ns for each system. The  starting geometry for each system was kept, but the initial vectors were randomly assigned to enrich conformational sampling. The  minimization procedure consisted of 2000 steps, involving 1000 steepest descent steps followed by 1000 steps of conjugate gradient energy minimization, with decreasing constraints on the protein backbone (500, 125 and 25 kcal·mol^−1^·Å^−2^) and a conjugate gradient minimization with no constraints. Next, the systems were gradually heated from 0 to 300 K over 20 ps using a Langevin thermostat with a collision frequency of 1.0 ps^−1^ in periodic boundary conditions with constant volume. Equilibration stage was run using the periodic boundary conditions with constant pressure for 1 ns (10 ns in the case of caspase-3 and factor Xa structures to ensure proper equilibration) with 1 fs step using Langevin dynamics with a frequency collision of 1 ps^−1^ to maintain temperature. Production stage was run for 50 ns with a 2 fs time step using Langevin dynamics with a collision frequency of 1 ps^−1^ to maintain constant temperature. Long-range electrostatic interactions were treated using the particle mesh Ewald method with a non-bonded cut-off of 10 Å and the SHAKE algorithm. The  coordinates were saved at an interval of 1 ps. The  computation of the maximum available volume (MAV) is given in the Appendix A.

### 3.5. Water Molecules Tracking, Hot-Spots Identification

AQUA-DUCT 1.0 software [56] was used to track water molecules for all proteases in each simulation replica. Tracking of water molecules was conducted in two specific regions: the *Object*, which represents the cavity of a particular interest, and the *Scope* representing the whole macromolecule. The *Object* was defined as a 4 Å sphere around the centroid of the active site residues of each protein (catalytic residues listed in Table 2), and the *Scope* was defined as the interior of a convex hull of α-carbon atoms in all structures. AQUA-DUCT was also used for identification of hot-spots, defined as the regions of the highest density of traced molecules within the protein interior. AQUA-DUCT is able to calculate hot-spots using two types of data: (i) using only the pathways of those molecules that entered the *Object* to calculate local hot-spots, and (ii) using the pathways of all molecules that entered the *Scope* region to calculate the global hot-spots. Both types of hot-spots were calculated for each of the simulation replica, and then simplified using the hs_gsimplifier.py script. The hs_simplifier.py script was used to analyze the positions of all identified hot-spots and grouped those hot-spots which were located within a radius of 3 Å in the case of global hot-spots, and 2 Å in the case of local hot-spots. Then it provided the information about the particular simulation replica in which the hot-spot was identified. The  information was kept and the simplified hot-spots are colour-coded according to their occupation. Those which were the most common were colored dark red, and those which were rare are white.

### 3.6. Molecular Docking and Validation

To ensure a high accuracy in pose prediction, we determined a fitting structural ensemble for each target that was able reproduce to binding modes of a maximal portion of non-covalent cocrystallized ligands. We evaluated the Glide standard-precision (SP) [57] as well as the smina [58] docking protocol throughout this study. While default setting were retained for Glide including the grid generation, an exhaustiveness of 16, a cubic search space with a side length of 21 Å, as well as a random seed of 42 was configured for smina. The  centroids for the respective search spaces were determined by computing the mass center of the cocrystallized ligand. The  RMSD between the docked pose and the respective cocrystallized ligand was computed using the rmsd.py script that comes with Maestro after protein structure alignment. In the case of unsatisfying pose prediction, we created structural ensembles (Appendix A) by clustering representative structures of MD simulations as detailed in our previous work [4]. For each protein, the docking protocol combined with the structural ensemble correctly reproducing the highest number of cocrystallized ligands was selected to assess the potential to discriminate random decoy compounds from known binders. Actives for each target were collected from various sources including crystal structures, the literature, as well as the PubChem database [59]. The  respective decoy compounds were generated using the novel DUDE-Z web server [60] with SMILES strings as input. Since LigPrep frequently generated multiple plausible protonation states and stereoisomers of the decoys, we chose the best docking score against the ensemble of each isomer. Together with the results from the known binders, the docking scores were submitted to the Screening Explorer web server [61] to determine enrichment metrics including the maximal reachable enrichment as well as the ROC AUC. The  following selectivity assessment was conducted by using SARS-CoV-2 M^pro^ inhibitors as decoy compounds combined with known binders as actives. The  ROC AUC metric was again obtained from the Screening Explorer web server and the scores were compared in a histogram computed using the Matplotlib [62] python library. To compare the absolute values of the docking scores among the different proteins, smina was used to rescore the poses obtained from Glide SP docking in the case of SARS-CoV-2 M_pro_, caspase-3, and chymase. Lastly, we aimed on deducing the structural factors for selective binding by visual inspection of the binding modes.

### 3.7. MD and MM/GBSA Post-Processing

To obtain more confidence in the results from docking, we post-processed the docking poses with the MM/GBSA protocol. Using Desmond (v2019-1), we conducted 2 ns simulations of all 576 ligand-protein complexes with different targets. We used the OPLS_2005 force field in an NPT ensemble with a temperature of 310 K maintained by the Nose-Hoover thermostat and atmospheric pressure maintained by the Martyna-Tobias-Klein barostat. The  orthorhombic periodic boundary system was solvated with TIP3P water molecules. Long-range interactions were treated with the u-series algorithm [63] and short-range interactions were cut off at 9 Å, while bonds to hydrogen atoms were constrained with the M-SHAKE algorithm. The  default relaxation protcol in Desmond was applied before the production phase. Atomic coordinates were deposited in an interval of 20 ps and the thermal_mmgbsa.py script that comes with Maestro (v2019-4) was applied to obtain binding free energies of the last 10 frames of the simulations, which were averaged thereafter.

## 4. Conclusions

Due to the current COVID-19 pandemic and the lack of specific therapeutics, many small molecules that inhibit SARS-CoV-2 M^pro^ have been proposed. This work aims to support further development of these compounds in order to avoid safety issues due to off-target binding, which is one of the major reasons for late stage drug attrition. We addressed the concern of selectivity and off-target binding of 33 published, experimentally confirmed SARS-CoV-2 M^pro^ inhibitors by predicting their affinity toward eight different proteases and profiling their active sites regarding hydration site and small-molecule hot-spots. Even though the selected off-target proteins presented a low global sequence identity to SARS-CoV-2 M^pro^, their binding sites were considerably similar. This similarity could explain the predicted affinities of the SARS-CoV-2 M^pro^ inhibitors, which presented a considerable overlap with actives against chymase, UCHL1, and cathepsin G. Interestingly, inhibitors of prostasin displayed higher predicted binding affinities to SARS-CoV-2 M^pro^ than its native inhibitors. Refining the SARS-CoV-2 M^pro^ inhibitors may be necessary to achieve higher affinities toward their designated target, as well as to improve selectivity and thereby decrease off-target binding. Around one third of the investigated compounds presented medium to high potential for endocrine disruption, altered drug metabolism, or cardiac adverse events based on the prediction of binding affinities towards 16 well established anti-targets. Our work showed that, while there are many proposed SARS-CoV-2 M^pro^ inhibitors, they generally exhibit poor selectivity and may cause pharmacological undesired effects by off-target binding. Even though the panel of proteases share a comparably low sequence identity and different substrate specificity, enzymes such as cathepsin G, factor Xa, as well as UCHL1 could be relevant off-targets for novel antivirals. If experimental testing and compound optimization efforts will be guided to achieve selectivity over the suggested anti-targets, novel antivirals could have an improved safety profile.

## Figures and Tables

**Figure 1 ijms-22-02065-f001:**
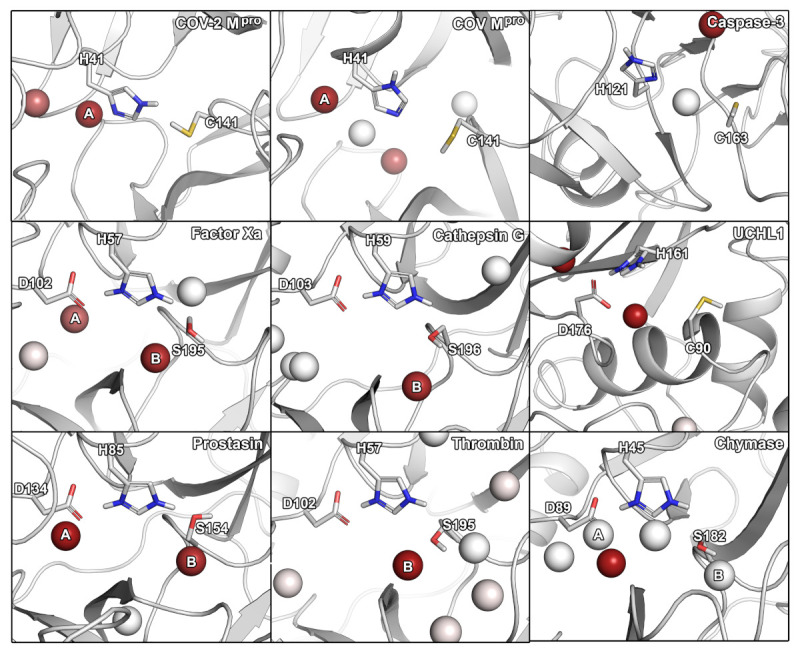
Hydration hot-spots of the selected panel of proteases in relation to the catalytic residues. To allow a direct comparison, the protein structures were aligned according to their catalytic residues. Two most consistently occurring hydration sites are indicated by A and B. The hot-spots are color-coded according to the occupancy of a particular region by identified hydration hot-spots. Hydration hot-spots with highest occupancy are colored in dark red, those with low occupancy in white.

**Figure 2 ijms-22-02065-f002:**
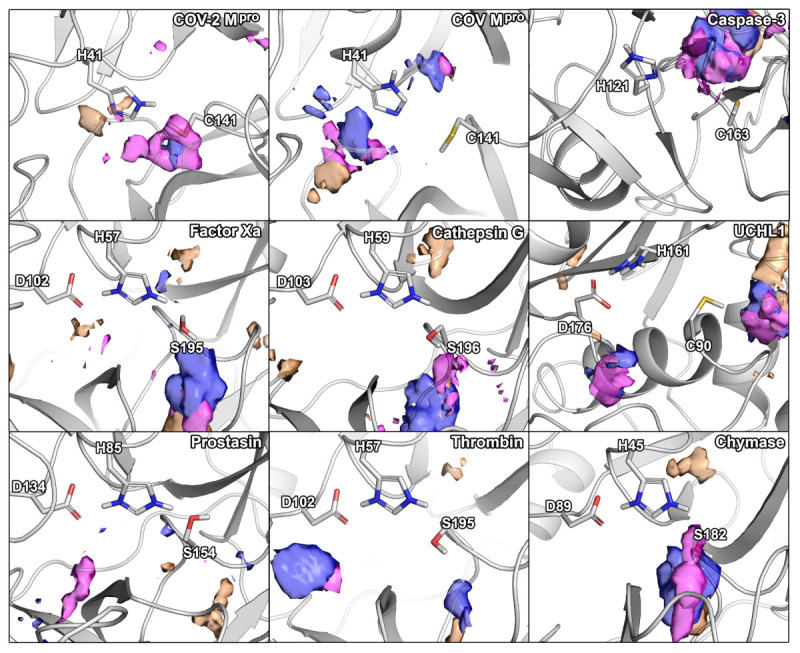
Small-molecule hot-spots of the selected panel of proteases in relation to their catalytic residues. Blue densities correspond to isopropanol, pink densities to pyridine, and orange densities to acetonitrile.

**Figure 3 ijms-22-02065-f003:**
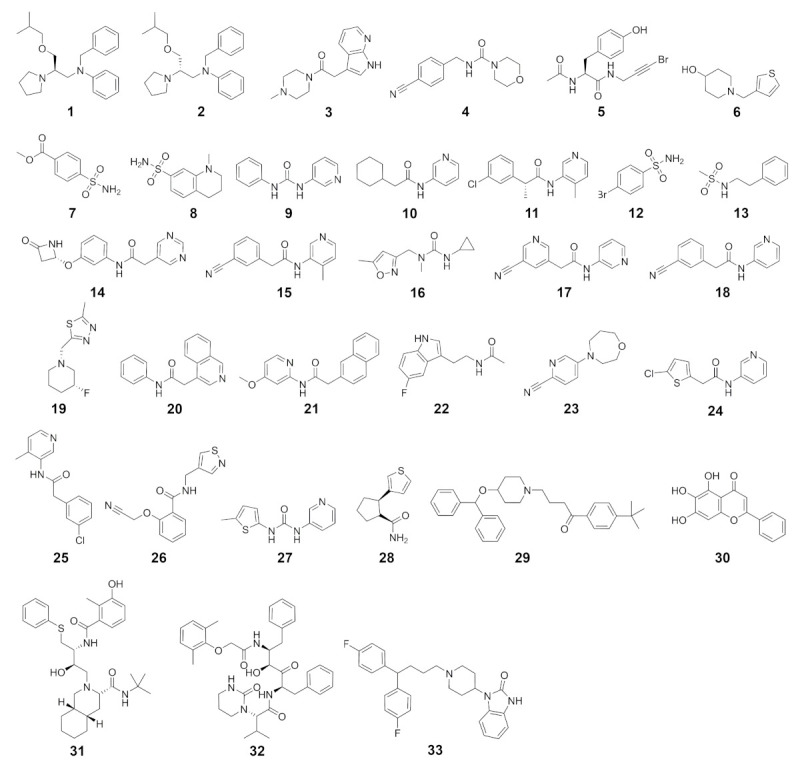
SARS-CoV-2 M^pro^ inhibitors considered in this study.

**Figure 4 ijms-22-02065-f004:**
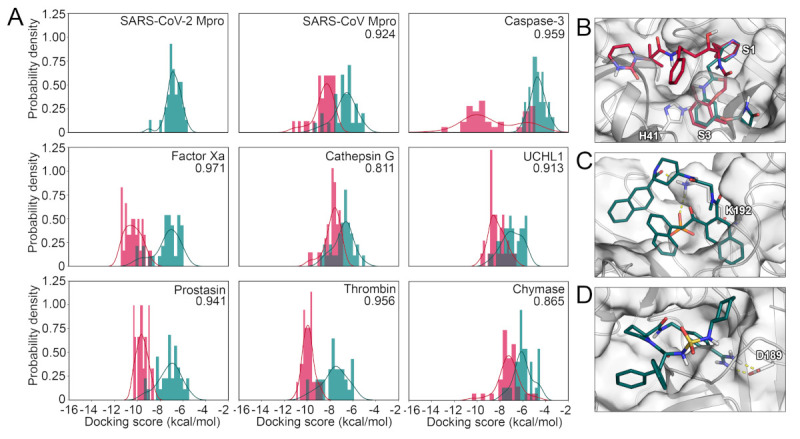
(**A**) Score distribution of SARS-CoV-2 M^pro^ inhibitors docked to the selected panel of proteases. The compounds designed against SARS-CoV-2 M^pro^ are shown in pine green, while the known actives for the remaining targets are shown in red. ROC AUC values with the SARS-CoV-2 M^pro^ inhibitors regarded as decoys for every target are shown. (**B**) Binding mode of compound **32** (red) and compound **14** (pine green) toward SARS-CoV-2 M^pro^. (**C**) Binding mode of compound **199** toward cathepsin G. (**D**) Binding mode of compound **177** toward thrombin.

**Figure 5 ijms-22-02065-f005:**
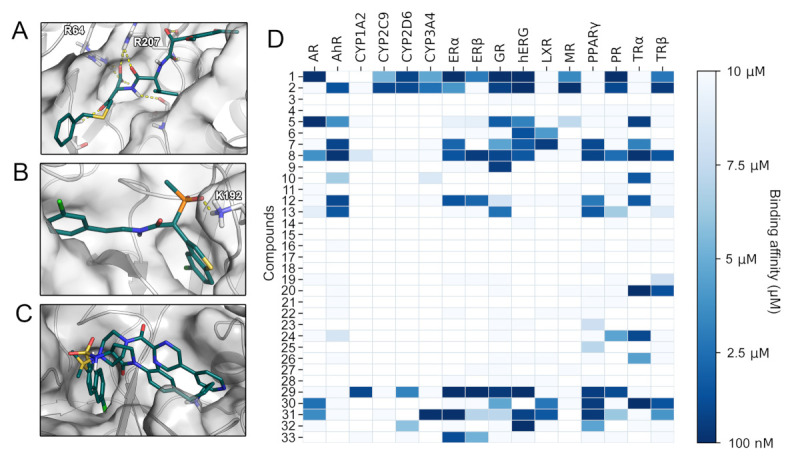
(**A**) Binding mode of compound **86** toward caspase-3. (**B**) Binding mode of compound **293** toward chymase. (**C**) Binding mode of compounds **130** and **137** toward factor Xa. (**D**) SARS-CoV-2 M^pro^ inhibitors examined with VTL. The predicted binding affinities of the assessed compounds **1-33** for 16 anti-targets are given.

**Table 1 ijms-22-02065-t001:** Proteins considered in this study.

Protein	Function	Anti-Target ^*a*^	Consequence of Inhibition
SARS-CoV-2 M^pro^	Viral replication	-	antiviral activity
SARS-CoV M^pro^	Viral replication	no	antiviral activity
Caspase-3	Apoptosis	yes	interference with development
Factor Xa	Coagulation	no	prevention of coagulopathies
Cathepsin G	Immune system	yes	interference with immune response
UCHL1	Protein degradation	yes	interference with development and homeostasis
Prostasin	Sodium balance	yes	alters homeostasis
Thrombin	Coagulation	no	prevention of coagulopathies
Chymase	Vasoconstriction	yes	interference with blood pressure

^*a*^ Description if the protein is regarded as anti-target.

**Table 2 ijms-22-02065-t002:** Similarity of proteins and active sites.

Protein	Catalytic	Global	AS RMSD	Site	Fold	Similarity
	Residues	Identity ^a^	(Å)^b^	Similarity ^c^		Index ^d^
SARS-CoV M^pro^	H41, C145	96.1%	0.2	0.91	α/β	0.90
Caspase-3	H121, C163	11.6%	1.9	0.67	α/β	−0.74
Factor Xa	H57, D102, S195	11.6 %	1.8	0.71	all-β	−0.67
Cathepsin G	H59, D103, S196	14.5%	1.8	0.61	all-β	−0.71
UCHL1	C90, H161, D176	15.7%	2.8	0.74	α/β	−0.98
Prostasin	H85, D134, S154	13.1%	1.6	0.69	all-β	0.26
Thrombin	H57, D102, S195	12.5%	1.8	0.74	all-β	0.31
Chymase	H45, D89, S182	19.0%	1.9	0.64	all-β	−0.49

^*a*^ Sequence identity of the catalytic unit to SARS-CoV-2 M^pro^; ^*b*^ RMSD between histidine, cysteine/serine, and aspartic acid residues relative to SARS-CoV-2 M^pro^;^*c*^ Similarity of the binding sites to SARS-CoV-2 M^pro^ determined by FuzCav; ^*d*^ Hodgkin similarity index of the binding sites comparison relative to the SARS-CoV-2 M^pro^ determined by PIPSA.

**Table 3 ijms-22-02065-t003:** Conclusions for selectivity of SARS-CoV-2 M^pro^ inhibitors.

Protein	Docking	Cosolvents	Hydration	Site Similarity ^a^
SARS-CoV M^pro^	**	*	**	***
Caspase-3	none	none	none	*
Factor Xa	**	**	*	*
Cathepsin G	***	*	none	*
UCHL1	***	*	*	none
Prostasin	**	none	*	**
Thrombin	**	none	*	**
Chymase	**	*	*	*

The potential for off-target binding of SARS-CoV-2M^pro^ inhibitors against the selected panel of proteases based on molecular docking, cosolvent MD simulations, and hydration site analysis. The potential was defined according to asterisks from none to three. ^a^ Consensus of binding site similarity determined with FuzCav and PIPSA.

## Data Availability

Raw data obtained from molecular docking, VTL, MM/GBSA, cosolvent MD, and conventional MD are available in a public repository on GitHub under https://github.com/mmodbasel/SARS-CoV2_selectivity, accessed on 4 January 2021.

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
