# Peer review of "Computational Selectivity Assessment of Protease Inhibitors against SARS-CoV-2"

_ijms, 2021, doi:10.3390/ijms22042065_

Round 1
Reviewer 1 Report
It is a nicely written article about using computational methods to access protease inhibitors against sars-cov2. I recommend publication of this article in IJMS.
Author Response
Thank you very much for the positive feedback!
Reviewer 2 Report
This is a well-done piece of work. The authors applied a series of computational approaches to assess the selectivity and potential toxicity of 33 reported SARS-CoV-2 Mpro inhibitors. In this work, the potential off-targets were chosen reasonably, the protocols for computational methods were validated using existing experimental data, the results were solid, and the discussion was insightful. I believe this work will be a useful guide for both academia and industry.
My major suggestion:
- Since there is no experimental result to support the conclusion in this work, which is understandable, I would suggest applying more expensive free energy calculations (e.g., MM-PBSA, QM-MM) to a subset of the docking results to further validate the robustness of the conclusion. But this is not absolutely required.
A few minor points:
- During the reviewing process, a few vaccines were already released. Please adjust the introduction accordingly.
- Please provide brief descriptions in Table 1 about why each protein is considered as an anti-target or not.
- Please consider explaining why these eight proteases were selected in the study early in the introduction.
- I am not sure how useful or easy-to-understand the binding-mode images in the figures are. I feel the authors could present the information in simpler 2D images.
Author Response
We would like to thank Reviewer 2 for constructive comments on our manuscript. We hope the changes alleviated all concerns.
Please, consult the attached file with all answers.
